



# Electrostatic discharge impacts on the main shaft bearings of wind turbines

Jian Zhao, Xiangdong Xu, Ola Carlson

Electrical Engineer Department, Chalmers University of Technology, Gothenburg city, 41258, Sweden

*Correspondence to*: Jian Zhao (zjian@chalmers.se)

**Abstract.** This paper studies the electrostatic discharge effect in wind turbines, a possible trigger source of main bearing current. A lab setup with a charge generator and downsized wind turbine was built to verify the impact of electrostatic discharge on the main bearing current. In the test, a fatal amplitude for the bearing current was found at only -93mV driven voltage on the shaft. Compared with the common mode voltage driven bearing current, the electrostatic discharge effect triggers the

bearing breakdown at a lower shaft voltage but much higher bearing current amplitude. The results demonstrate that the electrostatic discharge effect is a pattern of bearing current in wind turbines and is much more dangerous to the bearing.

## 1   Introduction

As economic development advances, more and more electrical power is demanded in day-to-day life and industrial production. However, traditional power generation methods are not usually environmentally friendly. Due to its renewability and almost

zero CO2 emissions, wind power is deemed a promising future power source.

Wind turbines only require an initial investment for construction and their maintenance costs are usually extremely low compared to the investment. However, replacing core components in the nacelle (located at the top of the tower) is extremely expensive. Of the core components, replacing the main shaft bearing (which holds the rotor weight and protects the shaft from the wind axial force) is very complicated and costly.

The main shaft bearing can break down for various reasons, such as poor fitting, poor lubrication, installation, material defects, excessive mechanical load and electrical current erosion (Radu, February 2010; ; and Weicker, 2021-09-28). Different failure patterns often trigger each other, with the bearing ultimately breaking down completely. One of the other failure patterns is bearing current erosion (Muetze, 2004; Muetze and Binder, 2005). This develops very slowly, making it difficult to know that a wind turbine's main bearing is eroding. Furthermore, although bearing current erosion will not burn out a bearing quickly, it

will worsen its operating state and trigger other bearing failure patterns.

The bearing failure mechanism under electrical current has been studied by different researchers (Muetze and Binder, 2007; Binder and Muetze, 2008; Radu, February 2010). However, most studies have focused on the bearing inside electrical machines, with very little attention paid to main shaft bearing current erosion. The electrical erosion process in a bearing is similar to that of a bearing current inside an electrical machine but its origins differ greatly. The studies of bearing current





sources in wind turbine main bearings mainly deal with the impact of common mode voltage (Haitham et al., 2014; Weicker, 2021).

The electrostatic discharge effect has been widely studied in the high voltage field (Mainra et al., 2022; Prashad, 2006) but its impact on a wind turbine's main shaft bearing has gone unnoticed by industry and academia. This paper analyses and investigates the electrostatic discharge effect on the main shaft bearing current. Its generation, transmission path and impact

on the main shaft bearing are studied and discussed in different sections. Various groups of tests were run in a laboratory setting to verify their impact on the main shaft bearing.

## 2    Electrostatic charge generation and accumulation in wind turbines

In nature, wind is produced by air movement arising from local temperature differences. Air contains a great many particles such as dust, water drops, ice and so on. When wind arises, these particles move with it and collide with each other. On a

microscopic level, the molecules of the various particles can exchange electron when they collide due to difference in their electron affinity. Thus, large number of free charges are generated and accumulated in the clouds contributing to atmospheric phenomena such as thunderstorms and lightning. When heavily charged clouds formed on top of a wind turbine, with a sharp blade tip, charges of opposite polarity, respect to the clouds charges, can be easily induce to the wind blades from the ground via its lightning protection system.

Additionally, the blads can be electrically charged when wind come into contact with the blads and separated. This effect is known as triboelectric charging. As shown in Figure 1, different material's electron affinity is shown.  Air is likely to given out electrons and polymers, such as epoxy and PTFE, have ability to attract electrons and become negatively charged.

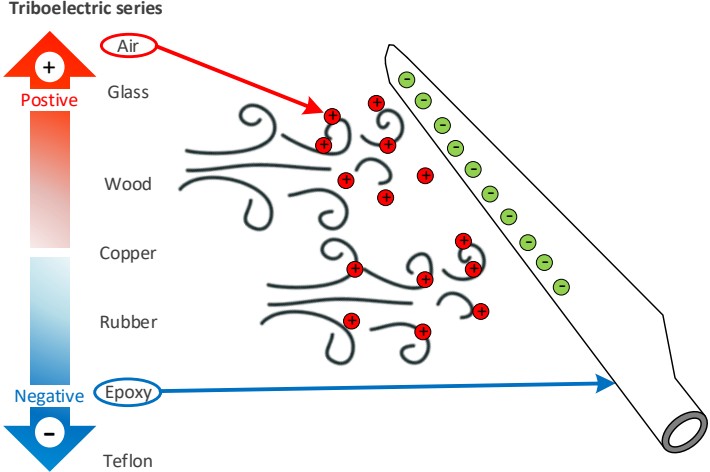

**Figure 1: Wind blowing over a wind turbine blade.**



One example of minimizing triboelectric effect in engineering application is the static dischargers on aeroplanes. An aeroplane's surface is often coated with polyurethane. As the plane moves in the air, its surface will accumulate charges and induce high electric field causing uncontrolled discharges and damage the aeroplane parts, especially modern electronics. To avoid such discharge risk, a number of needle-shaped dischargers are installed on the wings, as shown in Figure 2, these needles provide discharge points to dissipate accumulated charges back to the air via so called corona discharges.

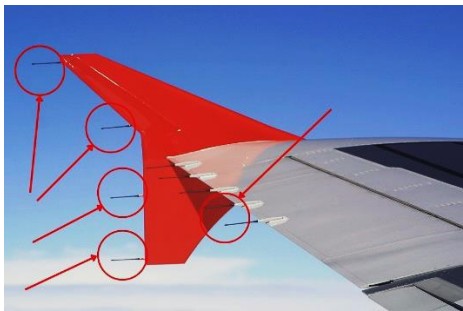

**Figure 2: Dischargers on aeroplane wings.**

### 3 Accumulated charges' path to ground

The accumulated charge on the wind turbine blades' polymeric dielectric surfaces builds up an electric potential and attempts to find its way to ground for dissipating the accumulated charge.

The structure of a typical wind turbine is shown in Figure 3. The blade has a multilayer structure, consisting of the blade frame, filling material, lightning protection components and surface coating. The frame, which provides the basic mechanical structure is usually made of wood, glass fibre, or carbon fibre. The outermost surface of the frame is coated with a layer of epoxy resin based polymeric coating to prevent erosion. Inside the blade, the lightning protection system comprises receptors and a copper net. The receptors that attract lightning from clouds are installed at the tip and middle of the blade respectively

(Méndez et al., 2018) and located on the outermost part of the coating. The copper net that provides a path to ground is embedded between the epoxy coating and the frame.

Regarding the blade structure, most components are dielectric materials, except for the lightning protection components. This provides a path for the surface-accumulated charges to be naturalized.

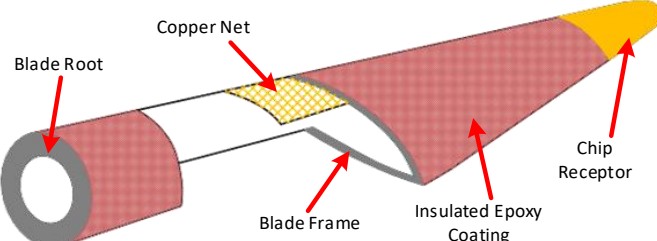

**Figure 3: Wind turbine blade structure diagram.**



At the wind turbine level, the continuously accumulated charges on the blades find a way to discharge to the wind turbine's lightning protection system and conduct to the ground (Hernández et al., 2019) resulting a current flow. There are different types of wind turbines configurations, with the most widely installed ones being geared and gearless turbines. Figure 4 shows a typical wind turbine with gearbox transmission, main bearing, and gearbox installed directly on the main shaft. The generator
is connected to the gearbox via an insulated coupling. In gearless wind turbines, the gearbox is eliminated, and the generator is installed directly with the main shaft.

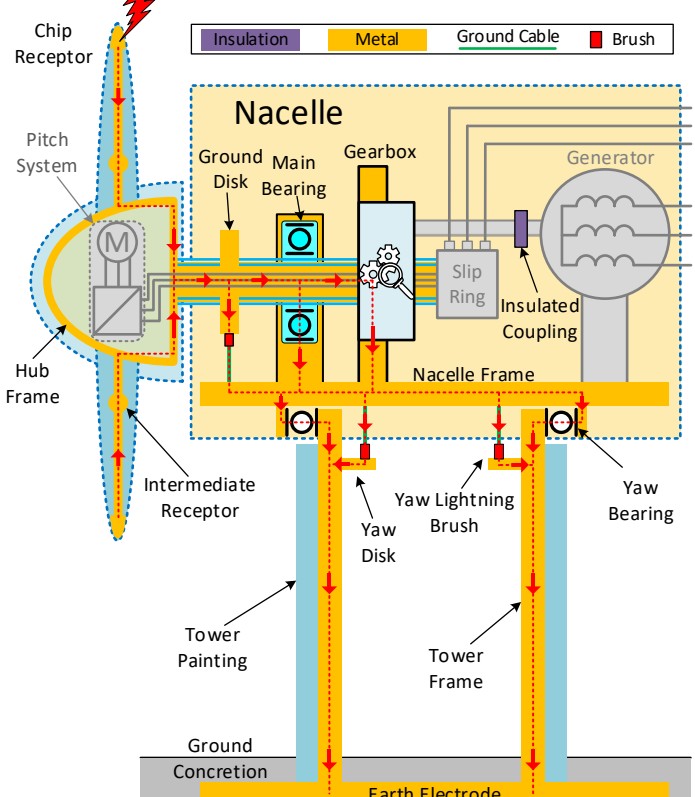

**Figure 4: A typical wind turbine lightning protection system.**

In the lightning protection system, the blades are electrically connected to the hub frame and main shaft. The main shaft is
fitted with a carbon brush which conducts charges on the main shaft to the nacelle frame and onward to the grounding system via the tower frame.

However, in reality, charges may take different paths from the main shaft to the nacelle frame. Though, the main shaft has a brush installed but there is also the main bearing and gearbox (geared turbine) or generator (gearless turbine). The main bearing and gearbox bearing or generator bearing provide a parallel current path for charges on the main shaft. The different paths are
indicated by dashed red lines with arrows in Figure 4.



If the electric current continuously goes through the bearing, the lubrication grease will be premature aged, and the races may get damaged. In the main shaft's ground path, the resistance of each component determines the amount of current on each parallel pathes.

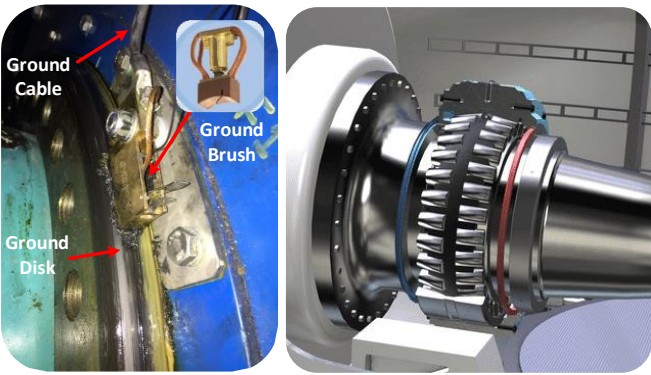

**Figure 5: Typical ground brush and wind turbine main bearing.**

Figure 5 shows a typical ground brush and a main shaft bearing. The ground brush is a cuboid block around 5cm long and 1 cm wide, usually made of carbon or metal with one of the cambered surfaces to match the surface of the ground disk. The main bearing is a double-row tapered roller bearing, comprising an inner race, outrace, rollers and lubrication grease. Its diameter ranges from a few dozen centimetres to meters. Thus, contact surfaces area of a main shaft bearing can be significantly

larger than the contact surface of a ground brush, resulting lower resistance path for charges to flow.

## 4    1 Bearing impedance model and bearing currents

Figure 6 shows a typical roller bearing structure. It consists of an inner ring, outer ring, rollers and roller cage, plus oil or grease packed between the races and rollers. In the turbine, the inner race's electrical contact is a mechanical connection to the main shaft, while the outrace is connected to ground via the bearing housing.

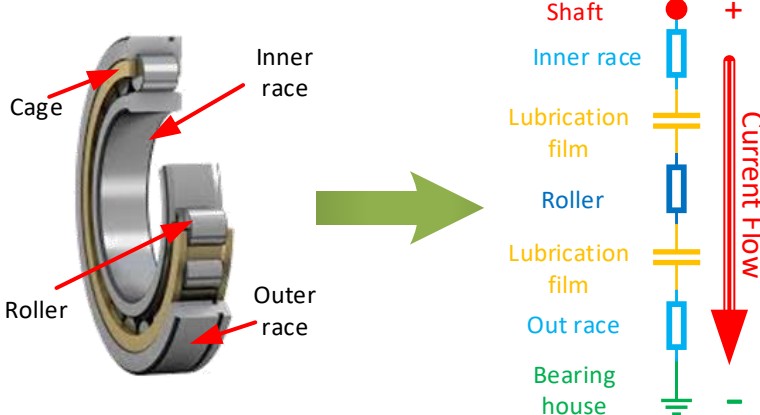




**Figure 6: Basic structure of a typical rolling bearing.**

The inner race, outer race, rollers and cage are made of metal. The rollers are held by the cage and are thus electrically connected. From the electric circuit standpoint, the bearing can be modelled as a capacitor and a resistor in series. Compared with the impedance of the capacitor, the resistance value is extremely small and usually disregarded in the circuit model.

The electrical properties of bearings have been studied by many researchers. In the rotation system, the bearing has different electrical states at different rotation speeds. At low rotation speeds, the main bearing supports the rotor, which weighs several tons. The lubrication grease between the roller and races is only partially filled and the bearing is operating in a conductive state. As the rotation speed of the rotor increases, a thin gap between the roller and races is formed and may filled with the lubrication grease, thus the bearing switches to its insulated state. A typical electrical model (Joshi, 2018) of the bearing is

shown in Figure 7. In this model, a switch is used to model the state transition of the bearing.

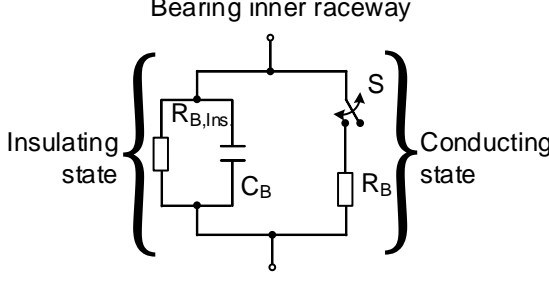

**Figure 7: Bearing electrical model under different states.**

## 5    Lab experiment setup

### 5.1   Electrostatic charges vs charge generation sources

Electrical charge flows around in the open air and there are always opportunities for it to attach to wind turbine blades. Due to the large size of wind turbines, the accumulated charge will generate a non-negligible electric potential on the main shaft. This electric potential tries to be neutralized to ground via a path of least resistance.

Above elaborated phenomenon needs to be verified and studied with simplified conductive paths in wind turbine, a downsized lab setup is therefore constructed. Furthermore, due to the turbine's smaller size, the charge accumulated on the lab scale wind

turbine blades is not sufficient to generate a sensible current flow and damages to the main bearing. So, an external discharge source is utilized to increase the charge density in the air.

To increase the charge density, a needle panel is used to release charge to the air by means of corona discharges. As shown in Figure 8, the needle panel is made of a steel net with number of needles welded onto it and connected to an adjustable negative high-voltage DC source. The panel is mounted in front of an electrical fan which provides a controllable air flow across the

needle panel and brings charges into a wind tunnel and increases the charge density inside.



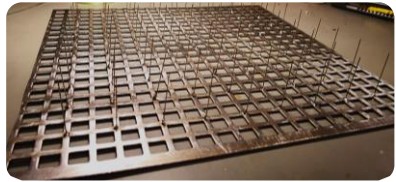

**Figure 8: Needle panel used to generate charges in air.**

## 5.2 Simplified wind turbine configuration

The basic structure of the wind turbine's lightning protection system is introduced in Session 3. In a real wind turbine, different
subsystems are mounted inside the nacelle and their layout differs greatly from turbine to turbine. In this study, the main point
of interest is the current in the main shaft bearing. Therefore, in the lab setup, the wind turbine configuration should be
simplified to highlight the charge-dissipating path and redundant subsystems are removed.

In a geared wind turbine, the gearbox is mechanically connected to the main shaft and to the generator via an insulated coupling.
Thus, the generator has no direct electrical contact to the main shaft. The main shaft voltage is dissipated via the ground brush,
main bearing and gearbox bearing.

In a gearless wind turbine, the main bearing is mounted between the main shaft and generator frame. The possible dissipation
paths for the main shaft voltage are the ground brush and the main bearing.

In one of the different types of gearless turbines, the main shaft is directly connected to the generator. The configuration is
similar to that of the geared wind turbine. The main shaft is grounded via the ground brush, main bearing and generator bearing.

To conclude the different transmission arrangements, the common grounding paths are the ground brush and main bearing. In
this study, the main topic of interest is the impact of electrostatic discharge on the main shaft bearing. Thus, the lab setup only
considers these two common paths. Figure 9 shows the simplified wind turbine configuration. It includes the charge generation
subsystem, lightning conducting subsystem, ground brush and main bearing of the turbine suspension system.

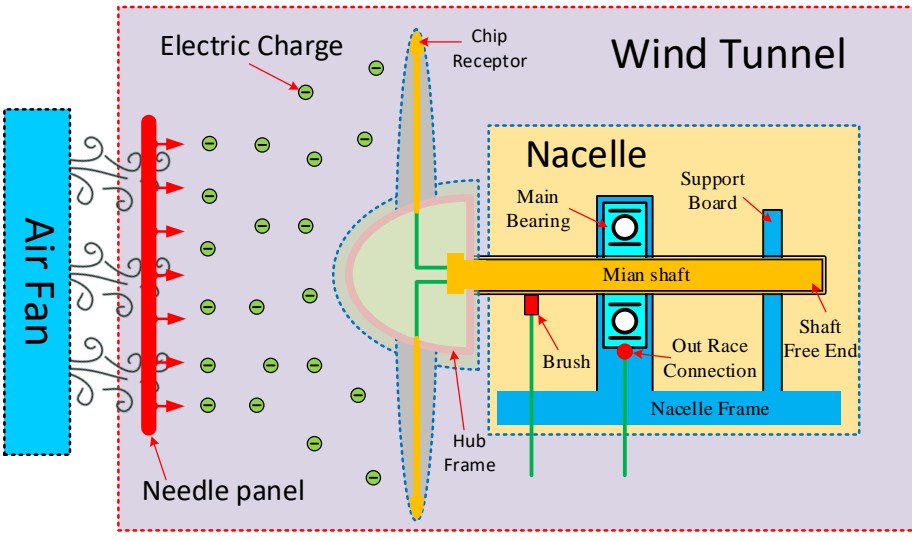



**Figure 9: illustration of the simplified wind turbine configuration utilized in lab study.**

## 5.3 Simplified wind turbine configuration

The downsized turbine part indicated in Figure 9 is located inside the wind tunnel which is made of wood and plastic sheet. Its centre height is around 1 m and its diameter about 75 cm. Most components in the downsized turbine are made of non-conductive plastic. This includes the rotor hub and blades, rotor suspension subsystem, lightning protection subsystem and

turbine frame.

The suspension subsystem supports the turbine rotor and locks the rotor from axial movement. Its principal components are the main bearings, nacelle frame and slip ring. Its configuration is shown in Figure 10. The suspension consists of a two-point system whose rotor is suspended by two bearings. The main bearing inside the bearing housing is a typical cylindrical roller bearing (SKF NU204) that can only support a radial load, while the slip ring bearing inside the slip ring is a typical ball bearing

that can support both radial and axial loads. The nacelle frame, bearing housing and mounting board are manufactured as a single unit. The slip ring is mounted on the mounting board, thus locking the slip ring in the axial direction. The main shaft goes through the main bearing and slip ring. At the rear end of the slip ring, the main shaft is locked together with the slip ring by screws. Thus, the rotor is locked in the axial direction by the slip ring and mounting board.

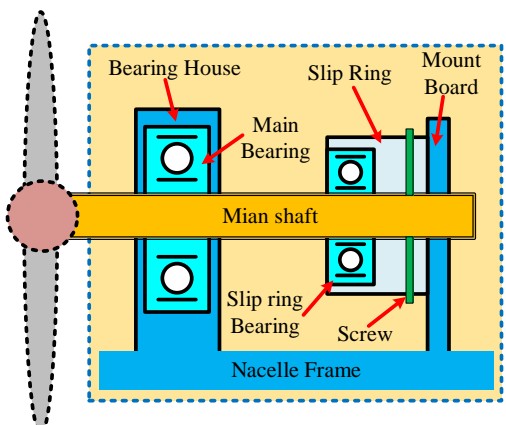

**Figure 10: Laboratory turbine suspension system.**

In the simplified setup, the lightning protection subsystem comprises coated plastic blades, plus the hub connection, main shaft and ground brush.



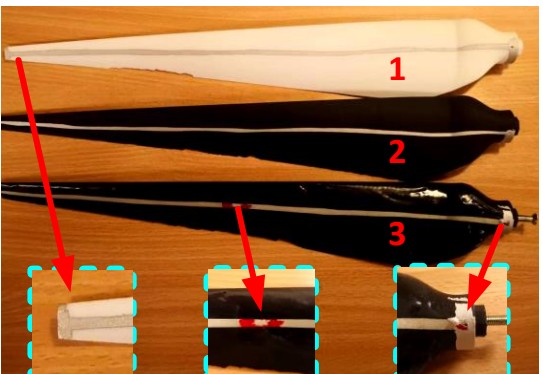

**Figure 11: Coated wind turbine blades.**

Figure 11 shows the coated blades. These are 3D printed in different colours of nylon. Electrically conductive silver lines are painted on both sides of the blades to provide the conductive path. Rectangular areas are painted onto the blade tips to simulate the chip receptors. A screw mounted at the root of each blade electrically connects to the silver painted lines via a copper wire. The whole surface of blade 3 in Figure 11 has been coated with a layer of epoxy resin, except for the tip and a pit in the middle as lightning receptors.

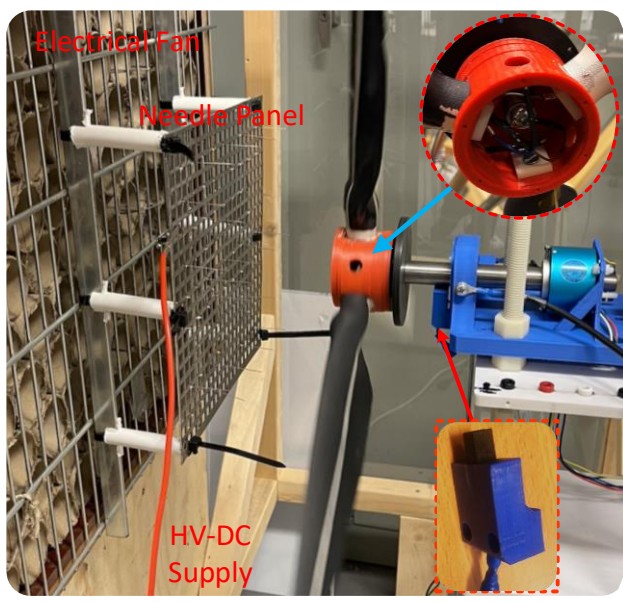


**Figure 12: ESD impact on the main shaft bearing, test setup.**

The main shaft is a hollow steel, with the hub mounted on the output side by a nut. Inside the hub, the screws into the blade's root are electrically connected to the shaft's rear end by three cables. These details also illustrated in the Figure 9. On the shaft's drive side (see sub-bottom figure in Figure 12), a carbon brush holder is mounted on the nacelle board and the carbon





brush is forced into contact with the surface of the main shaft by a spring. A conductive cable at the end of the brush connects the brush to the system's ground or test equipment.

## 6 Lab test and experiment results

To verify that the electrostatic discharge effect does impact the main shaft bearing, various groups of tests were designed for the downsized wind turbine lab system.

### 6.1 Lab test condition and bearing connection

According to the electrical model of the bearing introduced in Section 4, the bearing operates in the conductive state under low-speed rotation but switches to its insulated state as the speed increases. In a wind turbine, the main bearing normally operates in its insulated state. Under lab conditions, the selected bearing uses an oil based thin lubrication in the cylindrical roller, and thus often operates in its conductive state. To keep the lab turbine main bearing working state similar to the main

shaft bearing in wind turbine, grease based thicker lubrication was injected into the space between the rollers and races. The bearing with extra non-conductive grease is shown in Figure 13

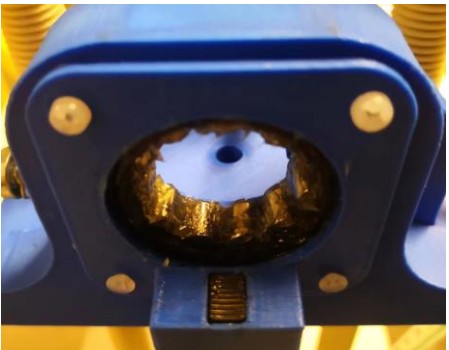

**Figure 13: Cylindrical roller bearing (in housing) packed with extra lubrication grease.**

To keep the test repeatable, a basic test condition was set for all tests in this study. On the charge generation side, the wind

speed was kept at a stable value (5-6m/s) and the voltage level was adjusted to achieve different charge densities in the wind tunnel. The turbine was driven by the wind at around 80 rpm and there was no load connected to the main shaft (in other words, the generator was disconnected from the shaft). The distance between the needle panel and turbine hub was fixed at 30 cm to limit the charge travel distance.





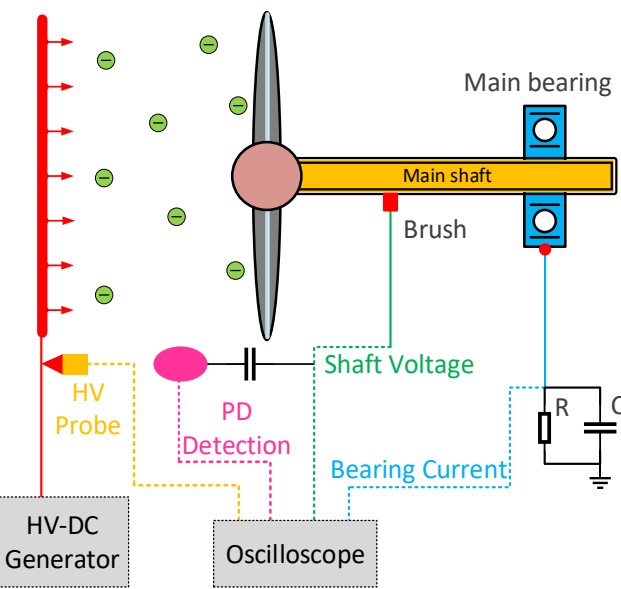

**Figure 14: Lab test connection.**

Focusing on the impact of ESD on the main bearing, the brush serves as a voltage observer (directly connected to the oscilloscope) and the main bearing's outer ring is grounded via a current shunt. Connected in this way all the current will, in principle, pass through the main bearing. Figure 14 shows the test connection for the setup.

Inside the bearing, the charge will accumulate at the roller and generate an unevenly distributed electric field within inusalted

bearing. Once the electric field strength is greater than the withstand strength of the lubrication grease at the bearing, an electric discharge will occur. A strong current will flow through grease and be conducted to ground via the outer ring and ground cable.

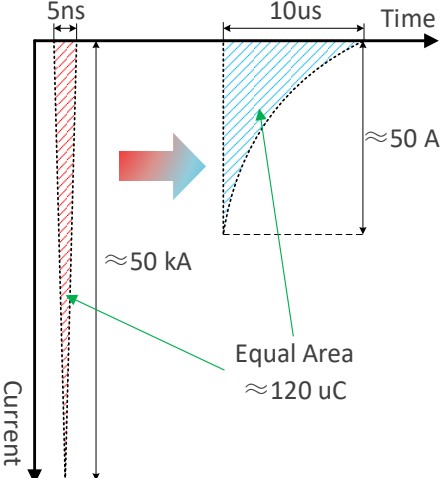

**Figure 15: bearing discharge current test method indication.**



The discharge has an extremely high amplitude current flowing for an extremely brief period. This is usually a matter of few nanoseconds with the current amplitude being in the kiloampere range. The amplitude and bandwidth of the discharge current are usually beyond the measurement range of an oscilloscope. To capture the current, the main bearing is grounded via a parallel resistor and capacitor (RC) circuit which measures the bearing discharge current. In this RC circuit, the capacitor slows the current flow, and the shunt resistor converts the current signal into a voltage signal. As shown in Figure 15, the RC circuit slows the progress of the discharge current and reduces its amplitude. But the total amount of dissipating charges remains the

same.

In the test setup, the used shunt capacitor is 100pF and the shunt resistor is 1kΩ. In the test connection shown in Figure 14, one side of the current shunt is connected to ground and the other feeds to the oscilloscope which measures the current passing through the main bearing as a voltage signal.

The input voltage on the needle panel is measured by the oscilloscope via a high-voltage probe. A high-pass filter is connected

to the shaft voltage signal and extracts the high-frequency discharge signal from it.

## 6.2  Induced shaft voltage due to charge accumulation

In the setup, different parameters influence the charge density in the wind tunnel. The main purpose of this study is to determine the charge's impact on the main bearing. Thus, the applied voltage to the discharge needles in the test is varied to adjust the charge density generated in the tunnel.

Test A - bearing state measurement:

Due to the extra grease injected into the main bearing, the bearing can operate in its insulated state. The first test verifies the bearing's operating state. Figure 16 shows the test connection; a 2.5V square wave voltage signal with frequency of 1k Hz is applied to the shaft via the carbon brush and the main bearing's outrace is connected directly to the oscilloscope to monitoring its voltage.

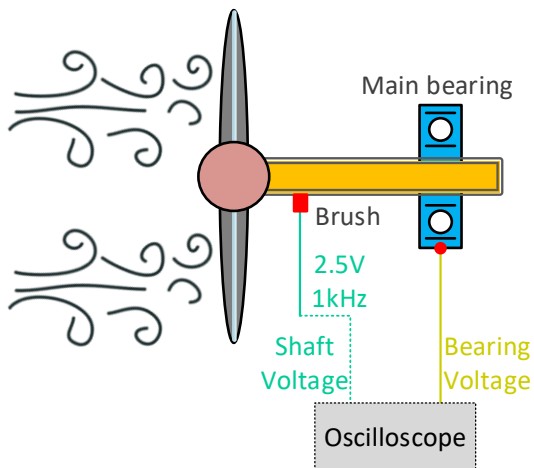


**Figure 16: Bearing state test connection.**



The turbine was rotated by the wind (5-6m/s) at around 80 rpm and no charge generation from the needle panel. The bearing, with and without grease, is shown in Figure 17 and Figure 18 respectively.

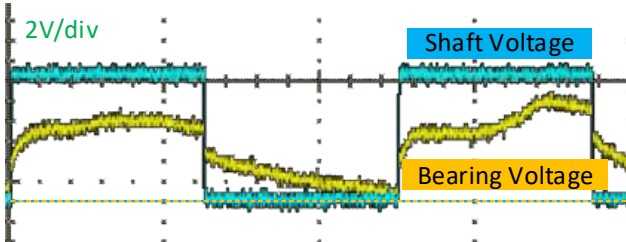

**Figure 17: Bearing voltage without grease.**

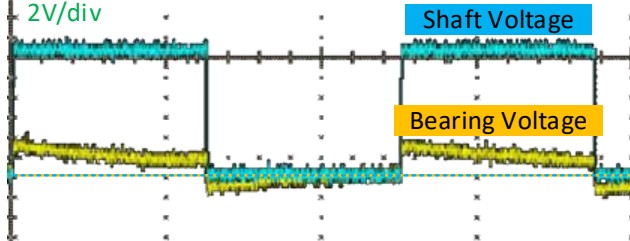

**Figure 18: Bearing voltage with grease.**

In the case of the bearing without grease, the bearing often operates in a conductive state, with the outer race voltage waveform nearly following the input voltage and showing very little voltage drop on the bearing impedance. In the case of the bearing

with grease, there was a significant potential difference between the inner and outer races, indicates an insulating states in bearing.

It is clear that injecting grease into the main bearing changes its operating condition from a conductive state to an insulated one. Furthermore, under the insulation state, there was no electrical breakdown in the bearing at 2.5V.

Test B – shaft-induced voltage

Using the test system connection in Figure 14, different DC voltage levels were applied to the needle panel. In the test, the charge accumulated / induced on the blade was conducted to the main shaft via the lightning protection system. The charges accumulated on the main shaft are impeded by the grease in the main bearing to the ground. The electrical properties of the main bearing mean that it acts as a dynamic capacitor, blocking DC but conducting AC. The charges from the air are unipolar particles, whose movement generates a negative DC current flow from the blade to the main shaft. These airborne charges are

stored on the bearing's inner race and build up a DC electric potential on the main shaft. As shown in Figure 19, the needle panel supplied with a voltage of -4kV in channel 1 and in channel 4, a voltage of -10mV is induced on the shaft. Because the airborne charges form negative sources, the voltage on the main shaft is also negative. Based on the bearing's current curve in channel 3, there is no current flowing through the bearing.



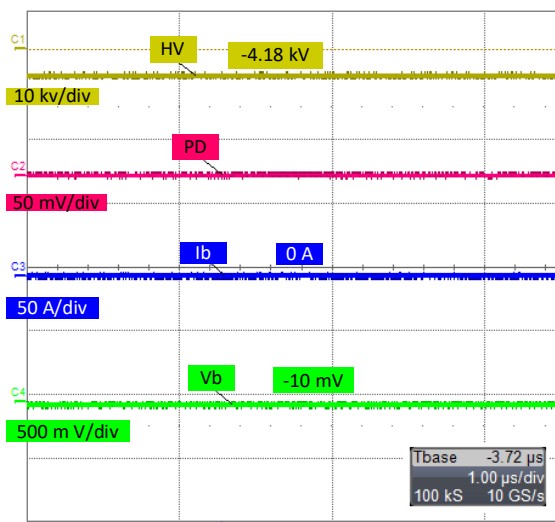

**Figure 19: Bearing voltage and current at -4.18 kV needle voltage.**

Test C – bearing breakdown

In Figure 20, with a -6.87kV DC voltage applied to the needle panel, the voltage build-up on the main shaft reached only -93 mV. From the bearing current curve, a breakdown clearly occurred on the bearing and its amplitude reached 62.9 A. During the breakdown process, the induced voltage on the shaft dipped to zero.

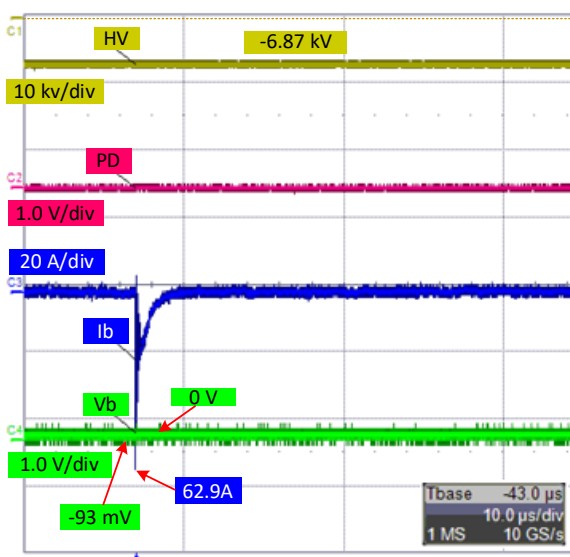

**Figure 20: Bearing voltage and current at -6.87 kV needle voltage.**

With the increasing needle voltage, the charge density in the tunnel further increases. When the needle voltage was increased to -14kV, the bearing broke down more frequently. The test results appear in Figure 21, in which the induced voltage reaches





-932mV. Within the same timespan as Test B, breakdown occurred three times, with the bearing current amplitude reaching hundreds of amps. During each breakdown, the shaft voltage dipped by several hundred millivolts.

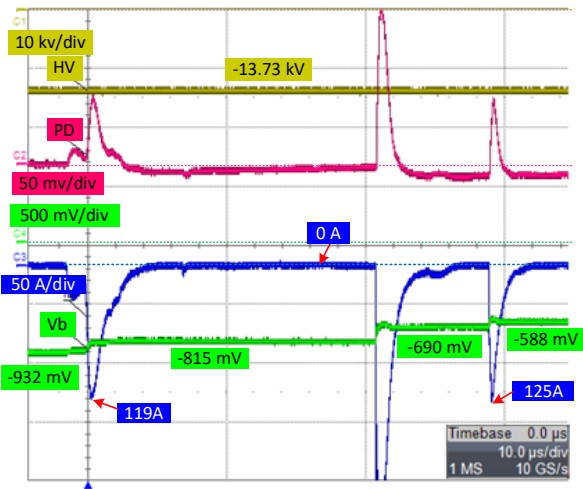

**Figure 21: Bearing voltage and current at -13.73 kV needle voltage.**

Test E – bearing conductive state change

Further increasing the charge density, the applied needle voltage was adjusted to -18.79 kV. As indicated in Figure 22, the
induced shaft voltage reached -4.83V. The breakdown occurred in the bearing during rotation, with the current flowing through the bearing peaking at 3.84kA. After the breakdown, the bearing's operating state reverted to a conductive state. There was a continuous current flow through the bearing, even after the voltage had dropped to nearly zero.

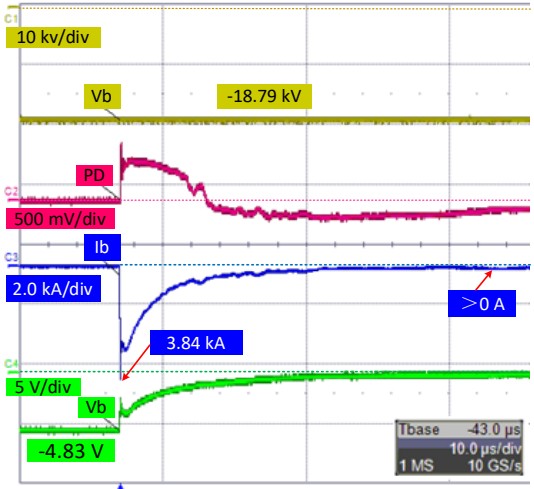

**Figure 22: Bearing voltage and current at -18.79kV needle voltage.**





## 7    Discussion of test results

In Test A, a 2.5V 1kHz AC voltage was applied to the bearing with grease lubrication, no breakdowns / discharges were observed. Comparing to the Tests C and D results, discharges occur more frequently with the shaft voltage was less than 1V, indicating discharges were largely related to charge accumulation at the bearing.

The bearing operated in its insulated state with the wind rotating the lab wind turbine. The grease thickness varies with rotation speed, the rollers' position and the load on the bearing and, from an electrical point of view, the bearing may be modelled as a capacitor. The capacitance varies with the thickness but generally, the capacitance stays within a stable range of dynamic values.

In Test A, a 1kHz AC voltage was applied to the shaft and the charges within the bearing capacitor were periodically polarized and depolarized. During each period, more charges did not have enough time to be accumulated and generated electric field that is strong enough to break down the grease dielectric.

Unlike the test A, the test B, a DC potential was built up at the shaft by means of charges accumulation via the blads. As shown in Figure 23 a, the accumulated charges built up a static electric field over the grease. But the amplitude of the electric field on the bearing grease is not strong enough to break down the bearing. Thus, the potential on the main shaft kept in a stable DC value, as shown in Figure 19.

In Test C and Test D, by increasing the voltage level to the needle panel, more charges were generated and travelled to the shaft via blades, and consequently, higher potentials at the shaft. With the rotation of the bearing, the grease thickness changed dynamically, and the electric field strength over the grease was inversely proportional to that thickness. In test C, even though the shaft voltage was only -93mV, the accumulated charges generate a strong enough local electric field to cause breakdowns in bearing when grease dynamically reached to a small enough thickness during rotating.

During the bearing breakdown, the electric arc is generated through the grease and causing a gas channel between roller and outerace. With increasing current flow, A plasma channel can be built (as shown in Figure 23 b) and the accumulated charges dissipated to the outrace, which is grounded via the mechanical connection. The shaft voltage dipped to zero and the electric field on the bearing grease was not enough to maintain the plasma channel. Thus, the plasma channel disappeared with the rotation and the breakdown process ended (Maradia and Wegener, 2015).




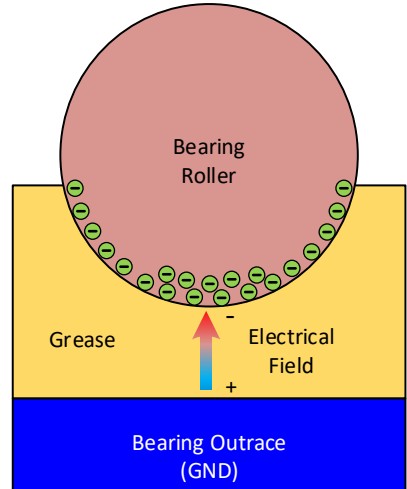 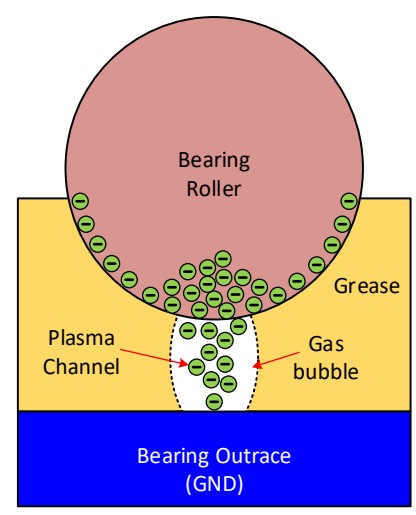


**Figure 23: Charge movement during breakdown.**

In Test E, with increasing charge density in the air, the charges were continuously conducted to the shaft. Once the electric field on the bearing was strong enough, breakdown occurred in the bearing grease. After the breakdown, the charges dissipated to ground via the discharge channel. With subsequent charges from the blades, these charges maintain a continuous flow of

current through the discharge channels inside the bearing grease. From the test results shown in Figure 22 it is obvious that, after the breakdown, the shaft voltage gradually dropped to nearly zero, but the current maintained to a stable value.

Comparing the test result from Tests C to E, the charge density increased gradually, and the shaft voltage increased from -93mV to -4.8V. It became easier and easier to trigger a breakdown in the bearing. The amplitude of the bearing current began appearing from 63A to 3.8kA. According to Figure 15, the current peak was suppressed by the shunt circuit. Assuming a

constant discharge duration of 5ns and the amplitude of the currents peak can be calculated as 1.2kA, 60kA, and 210kA in Tests C to E respectively.

## 8   Conclusions

This paper has studied and discussed the phenomenon of wind turbine main shaft bearing current. The ESD effect in wind turbines (a new pattern of bearing breakdown) is found in the main shaft bearing. This paper has analysed its root cause and

transmission path. Furthermore, to verify the analysis, a downsized wind turbine was built under lab conditions to simulate a real wind turbine. Several groups of tests were conducted on this lab turbine, with a fatal bearing current amplitude only found at a driven voltage of -93mV(Muetze, 2004; Muetze and Binder, 2006; Muetze et al., 2006). Compared with the AC voltage-driven bearing current, the ESD-driven bearing current appears at a lower trigger voltage and higher current amplitude. Furthermore, based on the charge movement in the test results, the breakdown mechanism of ESD-driven bearing current was

studied from a micro-perspective. The analysis and test results prove that the ESD effect in wind turbines is a non-negligible



source of main shaft bearing current in wind turbines and that it is much easier to trigger a fatal amplitude current in the main bearing.

## Acknowledgments

This study was carried out in the scope of the research project "Detecting and eliminating bearing currents for longer lifespan of main shaft bearings" funded by Swedish Energy Agency (Project ID: 2017-008071/44949-1). This project is launched together with SKF, Rabbalshede Kraft, Göteborg Energi AB, ABB AB and Skellefteå Kraft. The authors sincerely acknowledge their support.

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
