# Peer review of "Electrostatic discharge impacts on the main shaft bearings of wind turbines"

_Wind Energy Science, 2023_

## Author Comment (AC2)

Reviewer 2

The paper deals with an interesting topic - electrostatic discharge through bearings in wind turbines. Please consider the following comments:

- Please make it clearer when it comes to differences between the test setup and real wind turbine case. In a wind tubine (Fig. 4), an important point is the path of the discharge, either through the brushes or through the bearings. While in the test setup (Fig. 14), it seems that the only path is through the bearing. Is this correct? If yes, this must be highlighted more in the paper and conclusions.

Thank you for the valuable question. The connection is correct in the test. The explanation is given in next paragraph. In the paper, we add relevant explanation in session "6.1 lab test condition and bearing connection" (around line 200)

In the lab test, the brush is used as a test sensor for the shaft voltage. In the real wind turbine, based on our test, the brush impedance is > 10kΩ whereas the bearing is several ohms. They are not at the same level and not comparable. The figure below shows a test for the brush contact resistance in a 2MW wind turbine.

[Figure]

In the lab setup, the relative size of the bearing and the brush is not the same as the real wind turbine. The brush impedance is several ohms, and the bearing is about 10 kΩ. Thus, in the lab test, the brush is used as a test sensor instead of grounding path.

Updated content in the paper

"Different from the real wind turbine, in the lab test the brush is served as a voltage observer instead of a ground path. In the real wind turbine, based on our test, the brush impedance is > 10kΩ whereas the bearing is several ohms. They are not at the same level and not

comparable. In the lab setup, the relative size of the bearing and the brush is not the same as the real wind turbine. The brush impedance is several ohms, and the bearing is about 10 kΩ. Focusing on the impact of ESD on the main bearing, the brush only serves as a voltage observer (directly connected to the oscilloscope).The main bearing's outer ring is grounded via a current shunt. Connected in this way all the current will, in principle, pass through the main bearing. Figure 14 shows the test connection for the setup. "

- Please elaborate more on this sentence (page 17 - 312): "Compared with the AC voltagedriven bearing current, the ESD-driven bearing current appears at a lower trigger voltage and higher current amplitude". Please explain how this could be a conclusion of this paper.

Thank you for the reviewer's thoughts and comments, we have updated the paper, more discussion and explanation about this phenomenon are elaborated in session 7 around line 295.

Updated content in the paper

"In the AC-driven case in test A, the accumulated charge on the bearing appears as a tidal waveform, which means the charges accumulated and dissipated periodically from the source. In the ESD-driven bearing current, the voltage is a DC source, and the charge build-up field is a continuously increasing field. Compared with test A in Figure 18, even though the voltage level in test C (Figure 18) is lower, the partial electrical field is stronger than the case of AC voltage in test A. "

---

## Author Response (AR1)

Reviewer 1

In this paper, the influence of electrostatic effect on the main shaft bearing current of wind turbine is studied, and a laboratory device with charge generator and small wind turbine is established. The experimental results show that electrostatic discharge effect is a mode of bearing current of wind turbine, which is more harmful to bearings. The author needs to improve the article in the following aspects:

1 "Bearing current" is mentioned in the chapter title of Chapter 4, but the main body of this chapter only introduces the bearing impedance model and does not introduce the bearing current. Please make amendments to this problem.

Thanks for the reviewer's careful thoughts and comments, we have updated the title of Chapter 4 from "Bearing impedance model and bearing current "to" Bearing impedance model"

2 The titles of sections 5.2 and 5.3 are the same, please explain or modify them.

We apologize the type error in the manuscript, Title 5.3 updated from "simplified wind turbine configuration "to "5.3 Experimental setup "

3 In Figure 17, why is the measured bearing voltage waveform not a square wave? In addition, please explain the voltage difference between bearing voltage and shaft voltage.

Thanks for the comments, ideally, if bearing is in conductive states (i.e. bearing's outer race and inner race is electrically connected via balls), the bearing voltage should be identical as the shaft voltage. However, due to lubrication grease is a dielectric material, and with bearing rotation, bearing could become an insulating state, i.e. becomes an dynamic capacitor as elaborated in section 4.
Consequently, the voltage at the bearing outer race (bearing voltage) will be dynamically different from the shaft voltage, if without grease, due to rotating of the bearing, the bearing voltage is trying to follow the shaft voltage. If with grease, the small bearing voltage is induced from capacitive polarization and depolarization current in the dynamic bearing capacitor.

4 In Figure 21, why didn't the bearing voltage drop directly to 0V as in Figure. 22, when discharge breakdown occurred?

The bearing voltage drop is a dynamic procedure and caused by electrical discharges within the bearing, once the lubrication grease film breakdown, discharge appeared and the accumulated charge dissipated to the ground causing the voltage drop. But with the rotating, the dielectric grease film quickly builds up again, the newly generated charges within the wind blades and shaft system is not enough to sustain the discharges / breakdown the new dielectric films. Thus, discharges stochastically appeared in Figure 21.

In figure 22, by increasing the needle voltage (discharging rate increases), more charges are supply to the wind blades and thereafter to the bearing, with increased the charge density , the supplied current flow is able to maintain the breakdown of the lubrication film, thus the voltage dropped closed to zero and keep at a lower level.

5 What is the purpose of PD discharge detection? Will the increased capacitance of PD detection unit affect the detection of bearing voltage and current?

The PD test is used to identify the bearing breakdown instead of the bearing dynamic conductive state. The PD test circuit is a high pass filter, which pick up only the high frequency signal. The PD phenomenon is an extreme high frequency procedure that can be picked up by the PD detection circuit. The bearing dynamic conductive state is an instantaneous state of bearing that the roller touched the race caused by mechanical rotation. Compared with the breakdown, the dynamic conductive state is a longer procedure, but it is hard for us to identify it from the current waveform. Thus, we use PD detection signal to confirm the bearing film breakdown in the test.

The capacitor used for PD detection will influence the amplitude of the bearing breakdown voltage/current. But from the bearing aspect, the partial discharge phenomenon remains the same and the coupling capacitor's influence is limited. As illustrated in following tests:

2 additional test groups with and without PD test circuit connected were performed as the same condition as elaborated in chapter 6.2. Results shows the similar shaft voltage and bearing current results.

[Figure]

Figure 1 Lab test at 14,4kV needle voltage with and without PD detection system connected

[Figure]

Figure 2 Lab test at 13kV needle voltage with and without PD detection system connected

Reviewer 2

The paper deals with an interesting topic - electrostatic discharge through bearings in wind turbines. Please consider the following comments:

- Please make it clearer when it comes to differences between the test setup and real wind turbine case. In a wind tubine (Fig. 4), an important point is the path of the discharge, either through the brushes or through the bearings. While in the test setup (Fig. 14), it seems that the only path is through the bearing. Is this correct? If yes, this must be highlighted more in the paper and conclusions.

Thank you for the valuable question. The connection is correct in the test. The explanation is given in next paragraph. In the paper, we add relevant explanation in session "6.1 lab test condition and bearing connection" (around line 200)

In the lab test, the brush is used as a test sensor for the shaft voltage. In the real wind turbine, based on our test, the brush impedance is > 10kΩ whereas the bearing is several ohms. They are not at the same level and not comparable. The figure below shows a test for the brush contact resistance in a 2MW wind turbine.

[Figure]

In the lab setup, the relative size of the bearing and the brush is not the same as the real wind turbine. The brush impedance is several ohms, and the bearing is about 10 kΩ. Thus, in the lab test, the brush is used as a test sensor instead of grounding path.

Updated content in the paper

"Comparing to the field wind turbine, the brush used in the lab setup is served as a voltage probe and connect to the ground via the oscilloscope's 1 MΩ input impedance. To emulate a similar current path in the real wind turbine, the miniature bearing with a dynamic

impedance of 10 kΩ is selected by scaling. As based on our field measurements, due to lubrication and contaminations, the brush's dynamic contact resistance is generally high, over 10 kΩ, whereas the bearings' impedances are in the range of few ohms, thus current will mainly passing through the bearing instead of the brushes. In the lab setup, the miniature bearing's outer ring is grounded via a current shunt. Connected in this way current will mainly pass through the bearing. Figure 14 shows the test connection for the setup. "

- Please elaborate more on this sentence (page 17 - 312): "Compared with the AC voltagedriven bearing current, the ESD-driven bearing current appears at a lower trigger voltage and higher current amplitude". Please explain how this could be a conclusion of this paper.

Thank you for the reviewer's thoughts and comments, we have updated the paper, more discussion and explanation about this phenomenon are elaborated in session 7 around line 295.

Updated content in the paper

"In the AC-driven case in test A, the accumulated charge on the bearing appears as a tidal waveform, which means the charges accumulated and dissipated periodically from the source. In the ESD-driven bearing current, the voltage is a DC source and the charge build-up field is a continuously increasing field. Compared with test A in Figure 18, even though the voltage level in test C (Figure 20) is lower, the partial electrical field is stronger than the case of AC voltage in test A.

During the bearing breakdown, the electric arc is generated through the grease and causing a gas channel between roller and outer race. With increasing current flow, A plasma channel can be built (as shown in Figure 23 b) and the accumulated charges dissipated to the outer race, which is grounded via the mechanical connections. The shaft voltage dipped to zero and the electric field on the bearing grease was not enough to maintain the discharge channel. Thus, the discharge channel disappeared with the rotation and the breakdown process ended (Maradia and Wegener, 2015). "